# Discovery of RSV-Induced BRD4 Protein Interactions Using Native Immunoprecipitation and Parallel Accumulation—Serial Fragmentation (PASEF) Mass Spectrometry

**DOI:** 10.3390/v13030454

**Published:** 2021-03-11

**Authors:** Morgan Mann, David S. Roberts, Yanlong Zhu, Yi Li, Jia Zhou, Ying Ge, Allan R. Brasier

**Affiliations:** 1Department of Medicine, University of Wisconsin-Madison School of Medicine and Public Health (SMPH), Madison, WI 53705, USA; mwmann@wisc.edu; 2Department of Chemistry, University of Wisconsin-Madison, Madison, WI 53706, USA; dsroberts@wisc.edu (D.S.R.); ying.ge@wisc.edu (Y.G.); 3Human Proteomics Program, School of Medicine and Public Health, University of Wisconsin-Madison, Madison, WI 53705, USA; yzhu353@wisc.edu; 4Department of Cell and Regenerative Biology, University of Wisconsin-Madison, Madison, WI 53705, USA; 5Department of Pharmacology and Toxicology, University of Texas Medical Branch, Galveston, TX 77550, USA; yili4@UTMB.EDU (Y.L.); jizhou@UTMB.EDU (J.Z.); 6Institute for Clinical and Translational Research (ICTR), University of Wisconsin-Madison, Madison, WI 53705, USA

**Keywords:** RSV, BRD4, AP-MS, PPI, AP1, Wnt, innate immune response

## Abstract

Respiratory Syncytial Virus (RSV) causes severe inflammation and airway pathology in children and the elderly by infecting the epithelial cells of the upper and lower respiratory tract. RSV replication is sensed by intracellular pattern recognition receptors upstream of the IRF and NF-κB transcription factors. These proteins coordinate an innate inflammatory response via Bromodomain-containing protein 4 (BRD4), a protein that functions as a scaffold for unknown transcriptional regulators. To better understand the pleiotropic regulatory function of BRD4, we examine the BRD4 interactome and identify how RSV infection dynamically alters it. To accomplish these goals, we leverage native immunoprecipitation and Parallel Accumulation—Serial Fragmentation (PASEF) mass spectrometry to examine BRD4 complexes isolated from human alveolar epithelial cells in the absence or presence of RSV infection. In addition, we explore the role of BRD4’s acetyl-lysine binding bromodomains in mediating these interactions by using a highly selective competitive bromodomain inhibitor. We identify 101 proteins that are significantly enriched in the BRD4 complex and are responsive to both RSV-infection and BRD4 inhibition. These proteins are highly enriched in transcription factors and transcriptional coactivators. Among them, we identify members of the AP1 transcription factor complex, a complex important in innate signaling and cell stress responses. We independently confirm the BRD4/AP1 interaction in primary human small airway epithelial cells. We conclude that BRD4 recruits multiple transcription factors during RSV infection in a manner dependent on acetyl-lysine binding domain interactions. This data suggests that BRD4 recruits transcription factors to target its RNA processing complex to regulate gene expression in innate immunity and inflammation.

## 1. Introduction

Respiratory Syncytial Virus (RSV) is an enveloped, single-stranded, negative-sense RNA virus that infects ciliated epithelial cells in the respiratory tract [1]. RSV infection then spreads to the lower airways, where productive infection of small airway epithelial cells induces rapid activation of the innate immune response, resulting in secretion of pro-inflammatory cytokines [2], anti-viral interferons [3], exosomes [4], and damage-associated patterns [5] that mediate mucous production and leukocytic inflammation [4]. Worldwide, RSV infection is responsible for the plurality of pediatric hospitalizations in children age 5 and younger [6].

The innate immune response induced by RSV-infection is mediated by pattern recognition receptors (PRRs) such as the Toll-like receptors (e.g., TLR3) and Retinoic acid inducible gene (RIG-I) that monitor the airway lumen and cellular cytoplasm [3,7]. These PRRs recognize double-stranded RNA (dsRNA) and 5${}^{\prime }$-phosphorylated RNA as products of effective RSV replication and stimulate the nuclear translocation of the interferon regulatory factor IRF3 and the various nuclear factor kappa beta (NF-κB) transcription factors into the nucleus, where they cooperate to induce rapid expression of pro-inflammatory cytokines and anti-viral interferons [8,9,10].

Upon translocation to the nucleus, NF-κB RelA/p65 interacts with a protein called Bromodomain-containing protein 4 (BRD4), an epigenetic scaffold that mediates the RSV induced innate immune response [7,11,12,13]. BRD4 accomplishes this in part through its tandem bromodomains, which bind acetylated lysine residues on histones and transcription factors like RelA [8]. The resulting BRD4/RelA complex further interacts with Cyclin-dependent Kinase 9 (CDK9) to form the positive transcription elongation factor (p-TEFb). p-TEFb in turn initiates transcription via phosphorylation of RNA Polymerase II (Pol II) complexes paused at the promoters of NF-κB early-intermediate inflammatory genes [14]. BRD4-dependent transcriptional elongation involves the coordinated actions of histone acetyl transferases [15,16], cyclin dependent kinases [8] and ubiquitin ligases [17] that dissociate elongation factors and stimulate polymerase processivity. The mechanism by which the transcriptional elongation complex is reprogrammed in the innate immune response is not well understood.

In this study, we apply native immunoprecipitation (IP) of endogenous BRD4 complexes and high-resolution Online parallel accumulation-serial fragmentation mass spectrometry (PASEF-MS) [18] to address this knowledge gap by examining the BRD4 interactome and its dynamic changes in response to RSV infection. We also measure the dependence of these interactions on BRD4’s bromodomains by perturbing the complex with the small molecule bromodomain inhibitor, ZL0454 [19,20,21]. Our results demonstrate that BRD4 recruits several inflammation-modulating transcription factors during RSV infection, including ββ-catenin of the Wnt-signaling pathway and c-JUN (JUN) and Fos-Related Antigen 1 (FOSL1) of the AP1 complex. We further show that many of these transcription factors are sensitive to bromodomain inhibition. Both the interaction with AP1 and its sensitivity to bromodomain inhibition were confirmed in human small airway epithelial cells (hSAECs). We conclude that RSV dynamically enriches the BRD4 interactome for pro-inflammatory transcription factors, and that interactions are dependent on BRD4’s bromodomains.

## 2. Materials And Methods

### 2.1. Reagents & Chemicals

The 4-hexylphenylazosulfonate (Azo) used in these experiments was synthesized in-house as described previously [22,23]. The BRD4 selective BD competitive inhibitor, ZL0454, was synthesized as previously described [19,20] and determined to be >99% pure. All other reagents used for preparation of samples for MS analysis were ACS grade or higher and purchased from MilliporeSigma unless other-wise noted.

### 2.2. Virus Preparation And Infection

The human RSV long strain was grown in Hep-2 cells and prepared as described [7,24]. The viral titer of purified RSV pools was varied from 8 to 9 log PFU/ml, determined by a methylcellulose plaque assay [25,26]. Viral pools were aliquoted, quick-frozen on dry ice-ethanol, and stored at −70 °C until used.

### 2.3. Cell Culture and Treatment

A549 cells (human adenocarcinomic alveolar basal epithelial cells) were obtained from ATCC and grown in 10% FBS/F12K media (Corning, Corning, NY, USA). Primary human small airway epithelial cells (hSAECs) were immortalized using human Telomerase/CDK4 as previously described [27,28], and grown in SAGM small airway growth medium (Lonza, Walkersville, MD, USA). All cells were incubated at 37 °C, 5% CO${}_{2}$ until confluence.

A549 cells were washed twice with Phosphate-buffered Saline (PBS) and exchanged into serum-free F12K media prior to infection with RSV viral particles at a multiplicity of infection (MOI) of 1. Infected A549 cells were exchanged into FBS-containing media 2 h post-infection, and harvested at 24 h post-infection. hSAECs were stimulated with poly(I:C) (MilliporeSigma, Burlington, MA, USA, catalog no. P0913) by introducing the ligand to SAGM media at a final concentration of 50 µg/mL [29]. hSAECs were harvested four hours post-stimulation. The ZL0454 inhibitor was dissolved in DMSO and added to the relevant cell culture media at a final concentration of 10 µM. The ZL0454 inhibitor was added 18 h before infection/stimulation, and to the media during infection/stimulation.

### 2.4. Protein Extraction and BRD4 Immunoprecipitation

Cells were washed twice with cold PBS before lysis in 500 µL low ionic strength immunoprecipitation buffer (50 mM NaCl, 10 mM HEPES, 1% IGEPAL, 10% Glycerol) with 1 mM Dithiothreitol (DTT) and 1% protease inhibitor cocktail (MilliporeSigma, Burlington, MA, USA, catalog No. P8340) [30]. Lysates were sonicated three times for 10 s each time (BRANSON Sonifier 150, setting 4), and centrifuged for 5 min at 10,000 *g*, 4 °C. Approximately 3 mg of the supernatant was incubated overnight at 4 °C with 3 ug anti-BRD4 antibody (Cell Signaling, Danvers, MA, USA, catalog No. 13440) for BRD4 immunoprecipitation. A nonspecific isotype control antibody (LSBio, Seattle, WA, USA, catalog no. LS-C149375) was used as a negative control. 30 µL of Protein A magnetic beads (Dynabeads, Invitrogen, Waltham, MA, USA) were added, and the samples were incubated on a rotating mixer for 1 h at 4 °C. The beads were then separated from the supernatant with a magnetic stand. The beads were washed three times in low ionic strength immunoprecipitation buffer, transferred to a new tube, and washed once more; samples meant for mass spectrometry were washed three additional times in 50 mM NaCl/10 mM HEPES buffer before trypsin digestion.

### 2.5. Trypsin Digestion and Bottom-Up Sample Preparation

The magnetic beads were resuspended in 50 µL 0.2% 4-hexylphenylazosulfonate (Azo)/50 mM Ammonium Bicarbonate and reduced with 10 mM DTT at 37 °C for 1 h. Freshly prepared iodoacetamide solution (200 mM) was added to a final concentration of 20 mM, and the samples were incubated in the dark for 30 min. The beads were digested with 1 µg Trypsin Gold (Promega, Madison, WI, USA) overnight at 37 °C and agitation at 1000 rpm. The supernatants were collected, and the beads were further washed with 100 µL 0.1% Azo/50% Acetonitrile. The supernatants and washes were collected and dried in a vacuum centrifuge to remove the acetonitrile, and resuspended in 1% formic acid. The samples were then exposed to a mercury lamp (305 nm) for 5 min and centrifuged to degrade the Azo and remove byproduct salts. Finally, the samples were desalted using Pierce C18 tips (Thermo Scientific, Waltham, MA, USA) and resuspended in 0.1% Formic Acid.

### 2.6. Label-Free Quantitative Proteomics Analysis

Desalted peptides (2 µL) were loaded and separated on an IonOptiks Aurora UHPLC column with CSI fitting (Melbourne, Australia) at a flow rate of 0.4 µL/min and a linear gradient increasing from 0% to 17% mobile phase B (0.1% formic acid in acetonitrile) (mobile phase A: 0.1% formic acid in water) over 60 min; 17% to 25% from 60 to 90 min; 25% to 37% B from 90 to 100 min; 37% to 85% B from 100 min to 110 min; and a 10 min hold at 85% B before washing and returning to low organic conditions. The column directly integrated a nanoESI source for delivery of the samples to the mass spectrometer. MS spectra were captured with a Bruker timsTOF Pro quadrupole-time of flight (Q-TOF) mass spectrometer (Bruker Daltonics, Billerica, MA, USA) operating in PASEF mode, with 10 PASEF-MS/MS scans acquired per cycle. Precursors with charge states ranging from 0 to 5 were selected for fragmentation.

Protein identification and quantification were performed using MaxQuant v1.6.17.0 [31,32], with LFQ normalization restricted within sample groups. Four biological replicates with two technical replicates each were used for global label-free quantitation. Differential protein abundance was established using the “ProStar” and “DAPAR” [33] R packages for R version 4.0.3 [34]. In brief: protein abundance represented by MS signal intensity was ${\mathrm{Log}}_{2}$-transformed, and proteins were filtered to remove contaminants, reverse identifications, and proteins not quantified in at least 6 out of 8 technical replicates within at least one sample group. ${\mathrm{Log}}_{2}$ intensities were normalized to median of the global data set, and missing values were imputed via ssla for partially observed values within a condition, or set to the 2.5% quantile of observed intensities for observations that were missing entirely within a condition. A limma test was utilized to evaluate statistical significance based on an FDR-adjusted *p*-value of less than 0.05. For enrichment of protein in the BRD4 IP over the nonspecific IgG, a ${\mathrm{Log}}_{2}$ fold change of 1 or greater in either direction was used. For comparison of the BRD4 IP between treatments and biological conditions, the protein abundance was normalized to the abundance of BRD4 in the sample prior to analysis with ProStar and DAPAR, and a ${\mathrm{Log}}_{2}$ fold change threshold of 0.6 was chosen. P-values were FDR adjusted using the “p.adjust” function in base R (implementing the Benjamini-Hochberg procedure). Tandem MS spectra were visualized and plotted using Skyline [35] and the “ggplot2” [36] package for R version 4.0.3. Barplots for protein identifications were generated using the “ggpubr” [37] package for R version 4.0.3. Venn Diagrams were generated using the “vennDiagram” [38] and “RColorBrewer” [39] packages for R version 4.0.3.

### 2.7. Gene Ontology and String Analysis

Proteins identified in BRD4 IP samples but not in the nonspecific IgG pulldown were submitted to the online PANTHER tool for biological process gene ontology analysis [40,41]. STRING database analysis [42] was conducted using the list of putative BRD4 interacting proteins that simultaneously displayed (1) increased abundance on the BRD4 complex post-RSV infection and (2) reduced abundance on the RSV-stimulated BRD4 complex post treatment with the ZL0454 inhibitor. Molecular function gene ontology terms for these proteins were obtained through STRING’s built-in analysis functions. For significance, an FDR-adjusted *p*-value threshold was set at 0.05 for all analyses. GO Dot plots and UpSet plots were generated using the “ggplot2” and “ggupset” [43] packages for R version 4.0.3. Network images were generated in Cytoscape version 3.8.2 [44] using the “stringApp” [45] and “clusterMaker2” [46] plugins.

### 2.8. Western Blotting

Beads containing immunoprecipitated BRD4 complexes were suspended in 2x Laemli loading buffer and boiled at 95 °C for 5 min. The beads were then separated using a magnetic stand, and the supernatents were loaded on a 4–15% Criterion TGX Precast Protein Gel (Biorad, Hercules, CA, USA) for separation. Proteins were transferred to a nitrocellulose membrane using a Trans-Blot Turbo Transfer System (Biorad, Hercules, CA, USA) with a constant voltage of 25 V over 30 min. The membrane was blocked for 1 h using 5% milk powder in Tris-buffered Saline with 0.1% Tween-20 (TBST) and incubated overnight at 4 °C with anti-c-JUN (Cell Signaling, Danvers, MA, USA, catalog no. 9165) or anti-BRD4 antibodies (Cell Signaling, Danvers, MA, USA, catalog no. 13440) diluted 1:1000 in 5% milk powder/TBST. The membrane was washed thoroughly and incubated in VeriBlot IP Detection Reagent (Abcam, Cambridge, UK, catalog no. ab131366) diluted 1:200 in 5% milk powder/TBST. Imaging was performed via chemiluminescent detection using an Azure c500 gel imaging system (Azure Biosystems, Dublin, CA, USA), and densitometry was performed using FIJI version 1.53c [47].

## 3. Results

To investigate the effects of RSV infection on the interactome of BRD4, we utilized native immunoprecipitation and online PASEF-MS to quantify members of the BRD4 complex isolated from A549 human alveolar epithelial cells. A549 cells have been extensively utilized for analysis of airway innate responses, maintain characteristics of type II alveolar cells and are permissive for RSV infection [48,49]. A549 cells were infected with sucrose-cushion purified RSV (long) at a multiplicity of infection (MOI) of 1 for a total duration of 24 h before harvest. A subset of infected A549 cells were additionally treated with the BRD4 inhibitor ZL0454 (10 µM) beginning 12 h before infection. ZL0454 has been shown to effectively and specifically target the bromodomains of BRD4, with minimal cross-reactivity with other bromodomain proteins [19,21]. Control cells were treated with DMSO, and left uninfected. After 24 h of viral replication, all cells were harvested according to the workflow presented in Figure 1. Three combinations of biological and treatment conditions resulted: Control-DMSO (CD), RSV-DMSO (RD), and RSV-ZL0454 (RZ).

### 3.1. Identification of Putative BRD4 Interactors

Proteins were isolated from harvested cells via immunoprecipitation with an antibody specific to the long isoform of BRD4 (B). A nonspecific isotype control IgG (I) was used as a negative control to screen out contaminants. Accordingly, downstream analyses had 6 experimental groups to consider, resulting from the permutations of the biological conditions (“CD”, “RD”, “RZ”) and pulldown type (BRD4/“B”, IgG/“I”). After tryptic digestion, proteomic analysis, and removal of contaminants of reverse identifications, 2874 proteins were identified between all experimental groups. Notably fewer proteins were identified in the ZL0454 treated samples than in other experimental groups (Figure 2a). We speculate that many proteins and pulldown contaminants had diminished global abundance following the 42-h transcriptional blockade with ZL0454, resulting in fewer identifications in these groups.

After filtering to require identification in at least 6 (of 8) technical replicates in one or more biological conditions, this list was reduced to 2580 protein identifications. 1321 identifications were common to all sample groups, with another 302 identifications missing only in the “RZI” group, and 195 found in all but the “RZ” condition (Figure 2b). Relatively few (188) proteins were identified solely in BRD4 pulldown samples (i.e., not in IgG pulldown controls). These proteins were subjected to PANTHER gene ontology (GO) analysis, with an emphasis on biological process (Figure 2c). The selected proteins were highly enriched for transcriptional regulators and DNA binding proteins, which is consistent with the known roles of BRD4 in Pol II—dependent transcriptional initiation [11,50].

### 3.2. Quantitative Enrichment of BRD4 Interactors

Of the 2580 consensus protein identifications, 1603 were quantifiable in 6 or more technical replicates of at least one experimental group. Under these filters, sample groups demonstrated clear separation in principal component analysis (Figure 3a) with a high degree of Pearson correlation within sample groups (Figure 3b). This indicates good reproducibility within groups, as well as markedly different protein abundance profiles between them.

Proteins were evaluated as potential BRD4 interactors based on their relative enrichment in each BRD4 IP over their abundance in the matching IgG control group. (Figure 4). Using a ${\mathrm{Log}}_{2}$ fold change threshold of 1 and an adjusted *p*-value threshold of 0.05, we identified 230 proteins that were significantly enriched in the Control-DMSO comparison, 186 proteins that were significantly enriched in the RSV-DMSO comparison, and 416 proteins that were significantly enriched in the RSV-ZL0454 comparison. We speculate that the increased number of enriched proteins in the RSV-ZL condition stems from a reduced background protein abundance following ZL0454 transcriptional blockade. In support of this conclusion, we note that many proteins quantified in IgG pulldowns display differential abundance between biological conditions. Between all three comparisons, 557 unique proteins were significantly enriched and were categorized as potential interactors. Among these proteins, we identify ARID1A, and multiple subunits of the Pol II and Mediator complexes—all known interactors with BRD4 [30]. Additionally, we identify multiple transcription factors, ATPases, splicing factors, ribosomal proteins, and related eukaryotic initiation factors (Figure 4d). These enriched proteins are suggestive for additional roles of the BRD4 complex in mRNA processing and translation.

We note that the canonical BRD4 interactors, CDK9 and NF-κB RelA, are not flagged as potential interactors in this analysis, despite the fact that we identified both proteins during our experiment. In the case of RelA, we observed a high abundance of the protein in both the “RDI” and “RDB” sample groups, and accordingly enrichment could not be established (Appendix A). This likely reflects an increase in the protein’s global abundance that was captured on the nonspecific IP. CDK9, in contrast, was not reliably quantified (fewer than 6 intensity values in any given sample group) and thus was omitted from further analysis. Interestingly, we were able to reliably identify and quantify the enrichment of Cyclin-dependent Kinase 12 (CDK12) (Figure 5d & Appendix A), which functions in a highly similar manner to CDK9 [51,52].

### 3.3. RSV-Induced BRD4 Protein Interactions

Once we had identified these 557 potential BRD4 interactors, we proceeded to examine how RSV infection altered their relative abundance within the BRD4 complex. To correct for differences in BRD4 abundance by treatment, protein abundances in the pulldowns were normalized to that of BRD4. The resulting ${\mathrm{Log}}_{2}$(Protein/BRD4) ratios were then compared between the biological conditions to identify differential membership in the BRD4 complex. For this purpose, an absolute ${\mathrm{Log}}_{2}$ fold change threshold of 0.6 (corresponding to a positive 1.5 fold change in the usual scale) and an FDR-adjusted *p*-value threshold of 0.05 were set. Under these criteria, RSV infection resulted in a significant increase to the ${\mathrm{Log}}_{2}$(Protein/BRD4) ratio of 272 proteins (Figure 5a); 174 of these proteins displayed an absolute 2-fold change (${\mathrm{Log}}_{2}$ fold change > 1) in abundance or greater. In a manner similar to the list of putative interactors, we found that RSV-infection significantly increased the relative abundance of transcription factors and coactivators, ATPases, splicing factors, cytoskeleton binding proteins, and numerous proteins associated with translational machinery (Appendix A). In particular, we highlight the recruitment of multiple members of the AP1 transcription factor complex (i.e., c-JUN, FOSL1), as well as several associated proteins (e.g., MAP4K4, NACA) (Figure 6a). AP1 is a ubiquitous transcription factor that participates in the activation of pro-inflammatory cytokines [53,54,55]. Mitogen-activated protein kinase kinase kinase kinase 4 (MAP4K4) and Nascent polypeptide associated complex subunit alpha (NACA) also contribute to AP1 via activation of the Jun N-terminal kinase [56] and stabilization of c-JUN homodimers [57], respectively. This interaction suggests that BRD4 and AP1 cooperate to initiate transcription of pro-inflammatory cytokines.

In contrast to the proteins with increased relative abundance, only 35 proteins displayed a significant reduction (${\mathrm{Log}}_{2}$ fold change < −0.6) in membership to the BRD4 complex post RSV infection. These proteins were enriched for components of the nuclear pore complex and mRNA export machinery, as well as core cell cycle regulators (Appendix A).

### 3.4. Inhibitor-Sensitive BRD4 Interactors

The effect of bromodomain inhibition on the RSV-stimulated BRD4 complex was similarly determined by comparing the BRD4-normalized “RZB” and “RDB” groups (Figure 5b). In this contrast, we observed 81 proteins that significantly increased their membership in the BRD4 complex, as well as 160 that significantly decreased their relative abundance. 51 of these proteins displayed a ${\mathrm{Log}}_{2}$ fold change exceeding 1, and 103 displayed a ${\mathrm{Log}}_{2}$ fold change less than −1. Interestingly, we observed that treating A549 cells with 10 µM ZL0454 resulted in a reversal to many RSV-stimulated changes (Figure 5c); 95 proteins that had significantly increased abundance on the BRD4 complex following RSV-infection were significantly reduced following ZL0454 treatment. Six proteins displayed the opposite trend, with reduced abundance in the RSV-stimulated complex that was restored by bromodomain inhibition. This subset of differentially represented proteins is highly enriched in transcriptional coactivators, nuclear pore constituents, and mRNA splicing factors (Figure 7). This includes the AP1 complex member c-JUN, as well as the associated protein NACA. Notably, FOSL1 and MAP4K4 relative abundances were not significantly reduced by bromodomain inhibition, and c-JUN relative abundance remained above baseline, suggesting that BRD4 may interact with both AP1 c-JUN homodimers and AP1 JUN/FOSL1 heterodimers. CDK12 was likewise unaffected by bromodomain inhibition, which is consistent with the interaction mechanism of the related CDK9 protein, which interacts with BRD4 via its C-terminal domain rather than its bromodomains [58].

We also observe that BRD4 interacts with members of the E-cadherin (e.g., β-catenin/CTTNB1, γ-Catenin/JUP) complex in an RSV- and bromodomain inhibitor-specific manner. This complex is enriched in cell-cell junctions and regulates gene expression through direct and indirect mechanisms. β-catenin is known to indirectly interact with NF-κB RelA in a tissue and stimulus-specific manner that can either upregulate or downregulate NF-κB signalling. Similarly, β-catenin has been observed to interact with AP1 [59], and AP1 is known to cooperate with RelA [60]. This suggests that NF-κB, AP1, and Wnt signalling converge on BRD4 to regulate gene expression, and that bromodomain inhibition can interrupt this process.

### 3.5. Validation of the BRD4/AP1 Interaction

To confirm the discovered interaction with AP1 and its sensitivity to the small molecule bromodomain inhibitor ZL0454, we immunoprecipitated BRD4 complexes from human small airway epithelial cells (hSAECs). hSAECs are telomerase-immortalized human small airway cells that maintain stable epithelial morphology in monoculture, and reproduce both genomic and proteomic signatures of primary cells without early senescence [4]. hSAECs were treated with 10 µM ZL0454, and stimulated with the specific Toll-like Receptor 3 (TLR3) ligand poly(I:C) to induce inflammation similar to that of RSV infection [61,62]. Detection via western blot and a specific antibody for c-JUN confirmed that inflammation induces an interaction between BRD4 and the AP1 subunit c-JUN (Figure 6b). This experimental result also confirmed that the BRD4/c-JUN interaction could be partially disrupted by bromodomain inhibition with ZL0454.

## 4. Discussion

Respiratory syncytial virus is a common human pathogen and the single largest cause of pediatric hospitalization in the united states [6]. As a consequence of RSV replication in the lungs and airways, innate inflammation triggered in the airway epithelium plays a significant role in the progress and resolution of the disease. The innate inflammatory response proceeds largely through the actions of NF-κB [8] and IRF [7] transcription factors, which converge on BRD4, an epigenetic scaffold that interacts with transcription factors and cyclin-dependent kinases to enable Pol II-dependent transcriptional initiation. Previous works have identified over 250 high confidence members of the basal BRD4 interactome [30], including members of the SWI/SNF, DNA-directed Pol II complex, ribonucleoprotein, AP-2 adaptor, and spliceosomal complexes. Building on that foundation, we have utilized high-resolution, online PASEF-MS to identify 557 interactors over three biological conditions in A549 cells, and 272 interactors that are enriched on the RSV-stimulated BRD4 complex. These interactions are consistent with the previous body of work, and demonstrate that viral infection reorders the interactome of BRD4, enriching it for transcription and splicing factors, as well as kinases, ATPases, and translational machinery. We also demonstrate that a significant proportion of these interactions are sensitive to bromodomain inhibition, indicating that they are facilitated either directly or through downstream feedback signaling by BRD4’s acetyl-lysine binding activity. STRING network analysis and molecular function GO enrichment (Figure 7) indicate that these proteins are highly associated with nuclear pore function, RNA binding, and transcriptional coactivator activity.

### 4.1. Online PASEF-MS as a High-Resolution Tool for Dynamic Interactome Analysis

Mass spectrometry-based shotgun proteomics has been widely applied for the discovery of protein interactions via affinity purification samples [63]. While often considered the gold-standard for modern interactome studies, Affinity Purification Mass Spectrometry (AP-MS) often fails to identify low abundance interaction partners [64], despite the physiological importance that they can play. A variety of strategies have been implemented to improve quantitation of low-abundance peptides [65,66], but all come with trade-offs in the experimental workflow or the efficiency of protein identifications. In this study, we utilized online PASEF-MS [18], which implements trapped ion mobility spectrometry (TIMS) to simultaneously exclude singly-charged contaminant ions from MS acquisition, reduce chimeric spectra, separate isobaric precursors, and focus ions to improve sensitivity and throughput. This approach enabled us to deeply examine the interactome of endogenous BRD4, despite the low relative abundance of many interactors and the low cellular abundance of BRD4 itself.

Over three biological conditions, we quantified over 1600 proteins and validated 557 (35%) as potential BRD4 interactors. These proteins were identified with high-quality tandem mass spectra, and quantified based on technical duplicates of 4 biological replicates per biological condition and per antibody. Given our data’s consistency with previous works [30,67] and our independent validation of the BRD4/AP1 interaction in hSAECs, we are confident that the data presented is of high quality, and that our conclusions on RSV-stimulated and bromodomain-dependent interactions are valid.

### 4.2. BRD4 Recruits Inflammation-Modulating Transcription Factors during RSV Infection

During the process of viral infection, NF-κB RelA traffics to the nucleus and interacts with BRD4, CDK9 and other components of the positive transcription elongation factor (p-TEFb) [7]. The resulting complex phosphorylates Pol II, releasing it from a paused state on the promoters of pro-inflammatory and anti-viral genes [14]. This process results in a rapid and robust innate immune response which helps to reduce viral proliferation and recruit the cellular immune response. However, many NF-κB dependent genes experience delayed activation in response to inflammatory stimulus, and other genes with NF-κB binding sites remain inactive throughout the process [68,69,70], indicating complex transcriptional and epigenetic regulation. It is currently unclear how BRD4 circumvents these barriers.

To address this information gap, we conducted what is, to our knowledge, the first study to examine BRD4’s dynamic interactome in an unbiased manner. Previous studies have either focused on the basal interactome [30,67,71], or utilized highly-targeted assays to study individual interactions [72,73], such as the interaction between BRD4 and RelA [7]. This unbiased approach enabled us to discover novel, RSV-induced BRD4 interactions with transcription factors from multiple pathways. These included the AP1 transcription factor (e.g., c-JUN, FOSL1), and both β-catenin and Junction Plakoglobin, of the Wnt-signalling pathway. Other transcription factors are also enriched in this fashion, including DEAD-Box Helicase 3 (DDX3X), Metadherin (MTDH), and multiple members of the mediator complex (e.g., MED1, MED4, MED13, MED31). Remarkably, all of these proteins either modulate NF-κB [74,75,76,77] or directly facilitate the expression of pro-inflammatory cytokines and antiviral genes [55].

AP1 is a ubiquitous family of transcription factors composed of JUN-family homodimers and JUN/FOS-family heterodimers [78]. Multiple paralogs of both JUN and FOS exist and contribute to significant diversity in the effect and mechanisms of AP1 complexes. In the context of viral and TLR3-dependent inflammation, AP1 activates expression of numerous pro-inflammatory cytokines, including IL-6, IL-8, CD38, and TNF [55]. These genes are also highly induced via NF-κB, and interestingly, many NF-κB dependent genes also present AP1 binding sites and require the AP1 subunit c-JUN for efficient transcription [79].

Wnt-signaling through β-catenin is also associated with inflammation, but in a more nuanced fashion; Wnt/β-catenin can either repress or co-activate NF-κB-dependent inflammation via interactions with RelA in a cell-type and stimulus specific manner [80,81]. In some cases, NF-κB may serve as a coactivator for genes under the control of the Wnt pathway and drive aberrant expression of stem cell signature genes, which is observed to contribute to tumorigenesis [82]. Notably, interactions between β-catenin and RelA have been observed to be indirect; these two proteins did not physically associate in the absence of cell extracts [83]. Our results suggest that BRD4 may be a mediator of this association.

Finally, interactions have also been observed between β-catenin and AP1, and β-catenin binding sites are often enriched with AP1 binding sites [84,85]. c-JUN was reported to physically associate with β-catenin via interfaces at its N-terminus and DNA-binding domain [59]. Interestingly, this interaction promoted Cyclin D1 and c-Myc gene expression in a manner completely independent of AP1 binding sites. Altogether, our data suggests that the BRD4 complex integrates signals from the NF-κB, AP1, and Wnt pathways during RSV-infection and facilitates crosstalk between them.

### 4.3. RSV-Induced Interactions Are Bromodomain-Dependent

Small molecule bromodomain-and-extra-terminal (BET) inhibitors competitively occupy the bromodomains of BRD4 and related proteins, and have been used extensively over the last decade to disrupt bromodomain mediated transcriptional activity and probe related physiology [86,87]. In the context of airway inflammation, BET inhibitors have been demonstrated to block BRD4-mediated expression of pro-inflammatory cytokines and interferons, while preventing downstream airway remodeling via the epithelial-to-mesenchymal transition (EMT) [20]. Classically, BRD4 inhibitors suffer from low-specificity and dose-limiting toxicity [87]. In this study, we utilize the ZL0454 BET inhibitor discovered by our group [19]. ZL0454 was developed using structure-based drug design to identify chemistries that occupy the acetyl-lysine binding pocket of BRD4. Of these inhibitors, ZL0454 shows high selectivity for both bromodomains of BRD4. ZL0454 displaces acetylated lysine side chains from the bromodomain (BD)-1 and -2 of BRD4 with an ${\mathrm{IC}}_{50}$ of approximately 50 nM using time-resolved fluorescence resonance energy transfer (TR-FRET) assays. Comparing its selectivity, ZL0454 binds to BRD4 16-20 times higher than the BDs of closely related BRD-2, 3 and -T. In contrast to nonselective BET-isoform inhibitors, ZL0454 does not produce detectable toxicity in cell culture or in vivo [19]. Consequently, ZL0454 is a unique probe of acetylated lysine binding, enabling us to probe the bromodomain-dependence of numerous RSV-induced BRD4 interactions and provide insight into the mechanisms of BRD4-mediated innate inflammation.

Of the 227 BRD4 interactors that were recruited to the BRD4 complex during RSV-infection, we observe that 95 ( 42%) are disrupted to some degree by treatment with ZL0454, including most of the transcription factors described in the earlier section (e.g., c-JUN, CTNNB1, JUP, DDX3X, MTDH). In particular, we note that the recruitment of β-catenin and γ-catenin, as well as that of the mediator complex, is completely abrogated by competition for the acetyl lysine binding pocket of the BRD4 bromodomains. Considered together with the prominent role that β- and γ-catenin play in the EMT [88], this suggests that BET inhibitors may block airway remodeling by interfering with BRD4-mediated Wnt-signaling.

In addition to the catenins, we also observe partial depletion of members of the AP1 complex. In the case of c-JUN, we observe a nearly 3-fold reduction in response to bromodomain inhibition. Likewise, NACA—which is reported to stabilize c-JUN homodimers [57]—is completely displaced from the BRD4 complex by ZL0454. Despite this finding, we observed no reduction to the AP1 complex member FOSL1, with which c-JUN can alternatively form a heterodimer [78]. Further investigation will be required to confirm the exact nature of the BRD4/AP1 interaction, but these results suggest that BRD4 may recruit both AP1 homodimers and AP1 heterodimers via distinct interaction surfaces during RSV-infection.

### 4.4. Non-Transcriptional Roles of Dynamic BRD4 Interactors

Our discussion has focused primarily on RSV-induced protein interactors with transcriptional activity, but we also observe RSV-induced interactors with roles in other contexts, such as mRNA splicing (e.g., SF1, SF3A3, SF3B1, AKAP17A) [89,90], translation (e.g., RPS17, EIF3D, EIF5B) [91,92], protein folding (e.g., CCT5, CCT6A, CCT8) [93], and endoplasmic reticulum (ER)-targeting (e.g., SEC61A1, SEC61B) [94]. That is, we observe that BRD4 associates with protein complexes representing downstream elements of the central dogma of molecular biology. This, in turn, opens the possibility that BRD4 may funnel the products of activated genes into downstream processing complexes, thereby facilitating accelerated alternative splicing and translation, before transport to the ER for additional processing and secretion. In support of this, we note that alternative splicing can often be coupled to transcription [95], and splicing factors have been observed to interact with ribosomes and regulate translation [96,97]. While intriguing, BRD4’s role in these contexts has not been explored in the literature. Our data will therefore serve as a useful starting point for investigation into the potential post-transcriptional roles of BRD4.

## 5. Conclusions

In summary, using an unbiased affinity purification-mass spectrometry approach we identify dynamic, bromodomain-dependent changes to the BRD4 interactome in response to RSV infection. Taking advantage of a high-resolution mass spectrometer featuring a trapped ion mobility spectrometer front-end, we deeply interrogate the BRD4 interactome, identifying over 500 potential BRD4 interactors and over 270 RSV-induced changes to the interactome. This data will provide new hypotheses for understanding the pleiotropic role of BRD4 in the innate immune response and in viral infection.

## Figures and Tables

**Figure 1 viruses-13-00454-f001:**
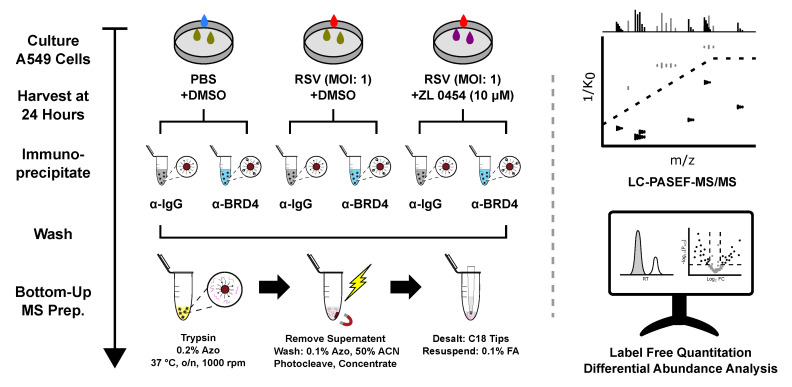
Affinity Purification Mass Spectrometry (AP-MS) experimental workflow, including sample preparation and data analysis. In brief, A549 cells were grown to confluence, treated with the bromodomain inhibitor ZL0454, and infected with RSV. Cells were harvested in a non-denaturing lysis buffer and BRD4 complexes were immunoprecipitated with anti-BRD4 or Isotype Control (IgG) antibodies and protein A magnetic beads. After extensive washing, the isolated proteins were digested on-bead using trypsin in a photocleavable surfactant (Azo) buffer [22,23]. After surfactant removal, the samples were desalted using C18 tips and analyzed via online LC-PASEF-MS/MS. Peptide identification and protein quantification were performed in MaxQuant. Differential abundance analysis was performed using DAPAR and Prostar.

**Figure 2 viruses-13-00454-f002:**
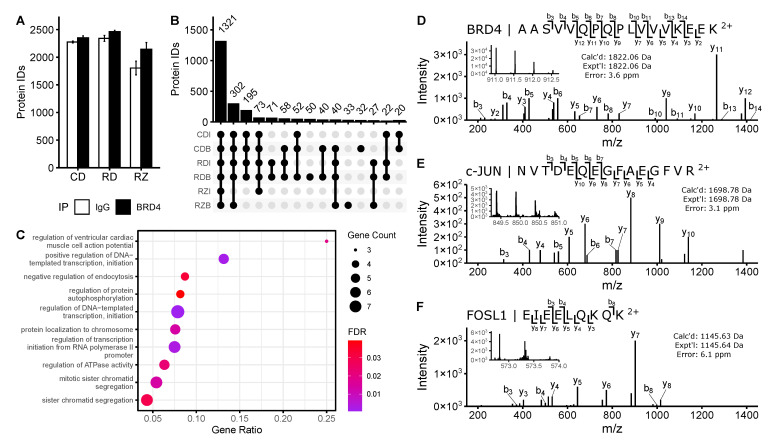
Identification of immunopurified proteins by online LC-PASEF-MS/MS. (**A**) Protein identifications per sample group. Bar plot represents the mean of n=8 biological and technical replicates. (**B**) UpSet plot of protein identifications shared between sample groups. Protein IDs were filtered to require n ≥ 6 identifications per sample group for representation. (**C**) PANTHER Biological Process GO analysis of proteins identified solely in BRD4 IP groups. Dot size represents the number of identified proteins within a GO group. Gene ratio indicates the fraction of all proteins within the GO group that were identified. Color represents the FDR-adjusted *p*-value of the GO over-representation test. (**D**–**F**) Tandem MS identification of characteristic peptides belonging to BRD4 and notable co-purified proteins. All fragment ion charge states are 1+.

**Figure 3 viruses-13-00454-f003:**
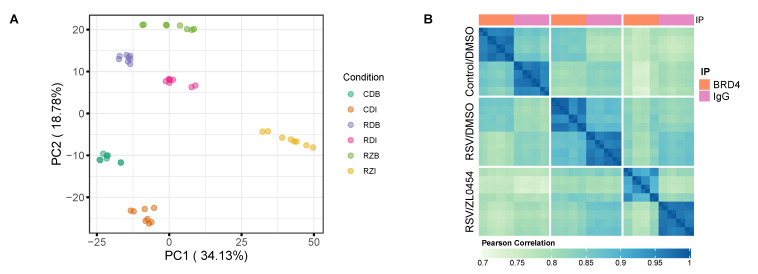
Sample reproducibility and inter-group variance. (**A**) Principal component analysis of per-sample ${\mathrm{Log}}_{2}$ protein abundances demonstrates good reproducibility within groups and separation between groups. (**B**) Heatmap with Pearson correlation scores between all samples (biological and technical replicates) in the experiment.

**Figure 4 viruses-13-00454-f004:**
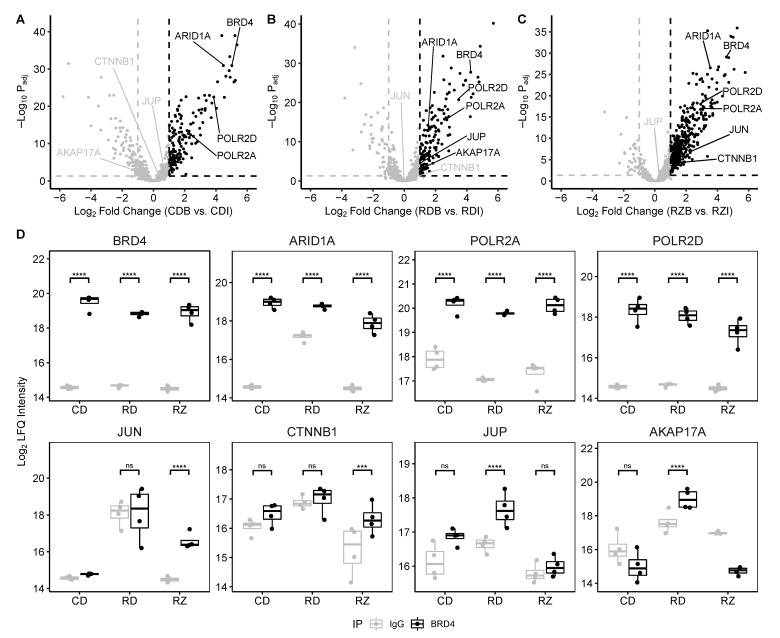
Potential BRD4 interactors are enriched over the IgG antibody. (**A**–**C**) Volcano plots demonstrating fold change enrichment of the BRD4 IP over the IgG IP in each biological condition. (**D**) Boxplots demonstrate enrichment of BRD4 and notable co-purified proteins in each biological condition. Boxplots represent n = 4 biological replicates per experimental group. Each data point represents the mean of n = 2 technical replicates. *** $P_{adj}<0.001$; **** $P_{adj}<0.0001$.

**Figure 5 viruses-13-00454-f005:**
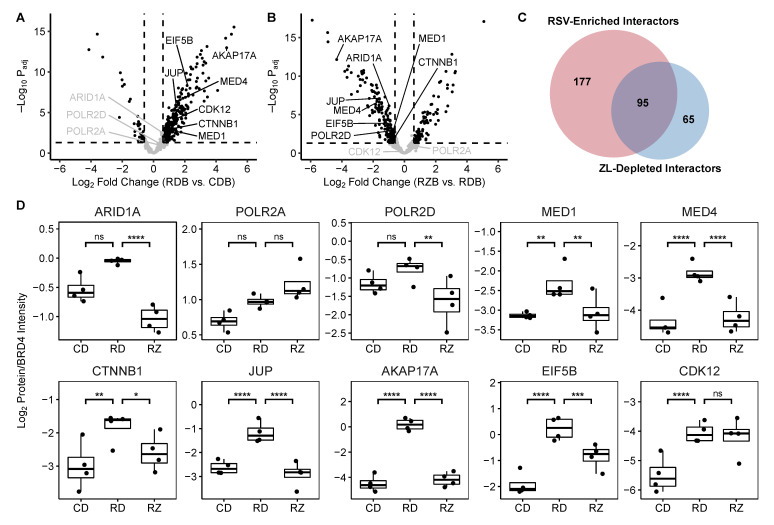
RSV-infection induces bromodomain-sensitive BRD4 interactions. (**A**) Volcano plot demonstrating fold change enrichment of BRD4 interactors induced by RSV-infection. (**B**) Volcano plot demonstrating fold change alterations to the RSV-stimulated BRD4 complex when treated with ZL0454. (**C**) Venn diagram illustrates overlap between RSV-enriched interactions and ZL0454-depleted interactions. (**D**) Boxplots demonstrate relative abundance of notable proteins in the BRD4 complex. Boxplots represent n = 4 biological replicates per experimental group. Each data point represents the mean of n = 2 technical replicates. * $P_{adj}<0.05$; ** $P_{adj}<0.01$; *** $P_{adj}<0.001$; **** $P_{adj}<0.0001$.

**Figure 6 viruses-13-00454-f006:**
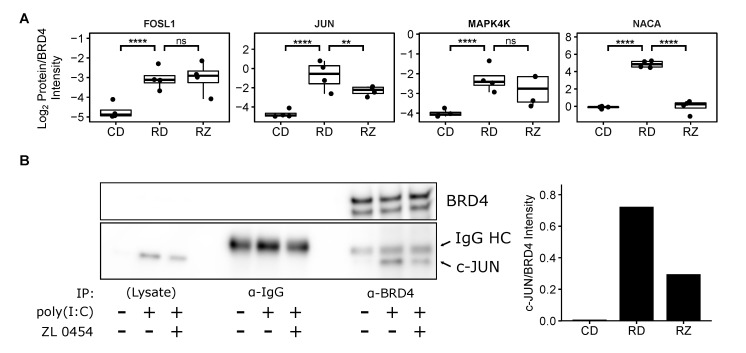
BRD4 recruits the AP1 transcription factor complex during RSV-infection. (**A**) Protein abundance boxplots of AP1 complex members and related proteins. Boxplots represent n = 4 biological replicates per experimental group. Each data point represents the mean of n = 2 technical replicates. ** $P_{adj}<0.01$; **** $P_{adj}<0.0001$. (**B**) Confirmatory western blot and densitometric quantitation, demonstrating that TLR3-induced inflammation in hSAECs triggers the BRD4/c-JUN interaction in a bromodomain-sensitive manner. (+/−) indicates the presence or absence of the respective compound in a given lane. “IgG HC” refers to the anti-BRD4 antibody heavy chain.

**Figure 7 viruses-13-00454-f007:**
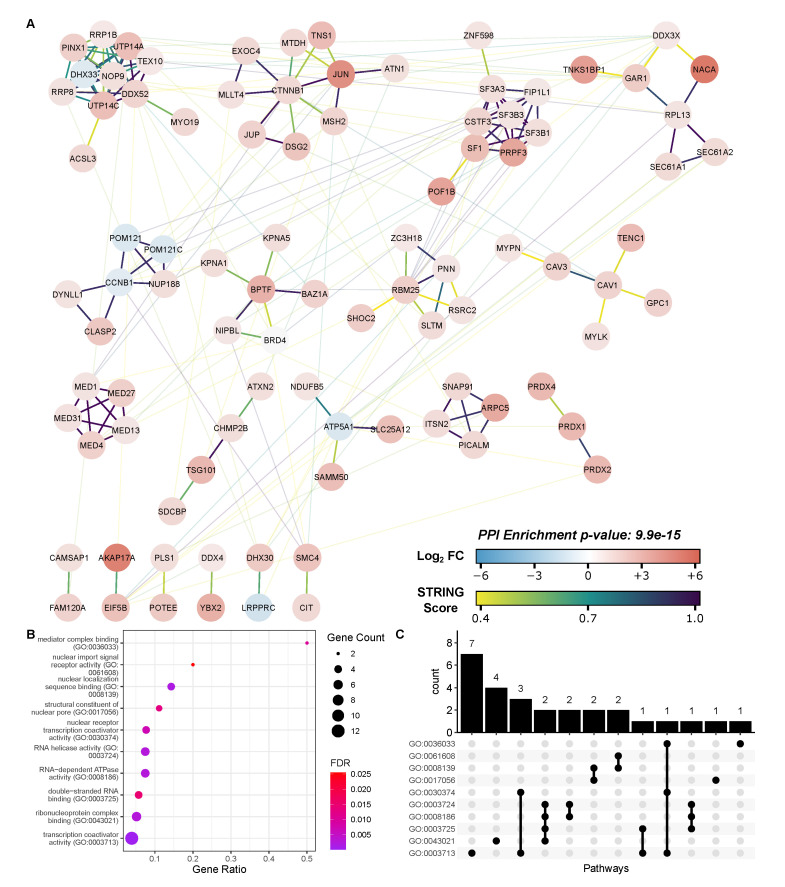
STRING analysis of RSV-inducible and bromodomain-sensitive BRD4 interactors. (**A**) Cytoscape network visualization of 101 RSV-inducible and bromodomain-dependent BRD4 interactors. Node color represents the RSV-inducible ${\mathrm{Log}}_{2}$ fold change. Edge color is keyed to the STRING score and represents interaction confidence. Low confidence interactions (STRING score < 0.4) and nodes without interactions are omitted. (**B**) STRING Molecular Function GO Analysis. Dot size represents the number of identified proteins within a GO group. Gene ratio indicates the fraction of all proteins within the GO group that were identified. Color represents the FDR-adjusted *p*-value of the GO over-representation test. (**C**) UpSet plot of proteins with shared GO terms. Each bar represents the number of proteins shared between marked pathways.

## Data Availability

All data contained within this manuscript are available upon reasonable request of the corresponding author. The mass spectrometry proteomics data have been deposited to the ProteomeXchange Consortium via the PRIDE [98] partner repository with the dataset identifier PXD023983.

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
