# Peer review of "Discovery of RSV-Induced BRD4 Protein Interactions Using Native Immunoprecipitation and Parallel Accumulation—Serial Fragmentation (PASEF) Mass Spectrometry"

_viruses, 2021, doi:10.3390/v13030454_

Round 1

Reviewer 1 Report

The results obtained by the authors were very interesting and excellent. I strongly hope that this research will continue to develop.

Author Response

Thank you for the kind comments on our study.  We agree that unique insights can be developed using this highly sensitive mass spectrometry technology, and that the experimental approach we have devised nicely complements the special Issue on Next Generation Technologies.

Reviewer 2 Report

Mann and colleagues utilized affinity purification–mass spectrometry approach to examine changes in the BRD4 interactome in response to RSV infection. Their data showed over 500 potential BRD4 interactors and suggested roles for BRD4-dependent immune response to RSV infection. Overall, the manuscript is straightforward and relatively easy to read. The authors appropriately referenced prior work demonstrating similar results.

Comments

For the BRD4/AP1 validation as shown on Figure 6B, the authors should include another control where cells were treated with ZL 0454 without poly(I:C) stimulation. The quantification of Western blot image should also be included in addition to an expanded explanation of bromodomain inhibition with ZL 0454.

Author Response

Thank you for the careful review of our manuscript and for the constructive comments.  We have considered these and revised the manuscript in the following manner:

  1. Please note that the AP1 signal from its association with BRD4 uninfected cells is below the limit of detection of the Western blot assay, so we do not feel that conducting a Western blot of the inhibitor-treated uninfected cells will yield interpretable results.  This question would require additional development of a more sensitive selective reaction monitoring assay, is beyond the scope of the current manuscript and would affect the interpretation of our study.
  2.  We have used densitometry to quantitate the Western immunoblot and present the results in the revised Fig. 6B.
  3. We have expanded the Discussion section to elaborate on the ZL 0454 compound.  Here we say :  "Classically, BRD4 inhibitors suffer from low specificity and dose-limiting toxicity [87]. In this study, we utilize the ZL 0454 BET inhibitor discovered by our group [19]. ZL 0454 was developed using structure-based drug design to identify chemistries that occupy the acetyl-lysine binding pocket of BRD4. Of these inhibitors, ZL 0454 shows high selectivity for both bromodomains of BRD4. ZL 0454 displaces acetylated lysine side chains from the bromodomain (BD)-1 and -2 of BRD4 with an IC50 of approximately 50 nM using time-resolved fluorescence resonance energy transfer (TR-FRET) assays. Comparing its selectivity, ZL 0454 binds to BRD4 16-20 times higher than the BDs of closely related BRD-2, 3 and -T. In contrast to nonselective BET-isoform inhibitors, ZL 0454 does not produce detectable toxicity in cell culture or in vivo [19]. Consequently, ZL 0454 is a unique probe of acetylated lysine binding, enabling us to probe the bromodomain-dependence of numerous RSV-induced BRD4 interactions and provide insight into the mechanisms of BRD4-mediated innate inflammation."